# FODMAP Fingerprinting of Bakery Products and Sourdoughs: Quantitative Assessment and Content Reduction through Fermentation

**DOI:** 10.3390/foods10040894

**Published:** 2021-04-19

**Authors:** Johannes Pitsch, Georg Sandner, Jakob Huemer, Maximilian Huemer, Stefan Huemer, Julian Weghuber

**Affiliations:** 1FFoQSI Austrian Competence Center for Feed and Food Quality, Safety and Innovation, Stelzhamerstrasse 23, 4600 Wels, Austria; johannes.pitsch@ffoqsi.at (J.P.); jakob.huemer@ffoqsi.at (J.H.); maximilian.huemer@ffoqsi.at (M.H.); 2Center of Excellence Food Technology and Nutrition, University of Applied Sciences Upper Austria, Stelzhamerstrasse 23, 4600 Wels, Austria; georg.sandner@fh-wels.at; 3Fischer Brot GmbH, Nebingerstraße 5, 4020 Linz, Austria; s.huemer@fischer-brot.at

**Keywords:** FODMAPs, fructooligosaccharides, charged aerosol detection, arabinose

## Abstract

Fermentable oligo-, di-, and monosaccharides and polyols (FODMAPs) are associated with digestive disorders and with diseases such as irritable bowel syndrome. In this study, we determined the FODMAP contents of bread, bakery products, and flour and assessed the effectiveness of sourdough fermentation for FODMAP reduction. The fermentation products were analyzed to determine the DP 2–7 and DP >7 fructooligosaccharide (FOS) content of rye and wheat sourdoughs. FOSs were reduced by *Acetobacter cerevisiae*, *Acetobacter okinawensis*, *Fructilactobacillus sanfranciscensis*, and *Leuconostoc citreum* to levels below those in rye (−81%; −97%) and wheat (−90%; −76%) flours. The fermentation temperature influenced the sourdough acetic acid to lactic acid ratios (4:1 at 4 °C; 1:1 at 10 °C). The rye sourdough contained high levels of beneficial arabinose (28.92 g/kg) and mannitol (20.82 g/kg). Our study contributes in-depth knowledge of low-temperature sourdough fermentation in terms of effective FODMAP reduction and concurrent production of desirable fermentation byproducts.

## 1. Introduction

### 1.1. Definition of FODMAPs 

The acronym FODMAPsstands for fermentable oligosaccharides, disaccharides, monosaccharides, and polyols and represents a group of short-chain carbohydrates, including fructose, fructans, fructooligosaccharides (FOSs), galactooligosaccharides (GOSs), galactans, lactose and the sugar alcohols maltitol, mannitol, sorbitol, and xylitol [1]. FOSs are considered fructose oligomers with a low degree of polymerization (approximately up to 12 units of fructose), in comparison with polymers, which are considered fructans [2].

### 1.2. FODMAP-Related Diseases 

FODMAPs have been determined to play a major role in gastrointestinal (GI) disorders because they are not absorbed properly in the gut. It is generally estimated that approximately 6–10% of the overall population is affected by functional bowel disorders; in the Western world, these disorders affect up to 20% of the population [3]. The pathophysiology of irritable bowel syndrome (IBS) is multifactorial, including altered brain–gut signaling, carbohydrate malabsorption, gut dysbiosis, immune dysregulation, and visceral hypersensitivity [4]. The FODMAP mode of action is connected to osmotic activity, forcing water into the GI tract. Furthermore, the sugar molecules are fermented by intestinal bacteria, which increase hydrogen, methane, and carbon dioxide production in the GI tract. Thus, the elevated luminal distension leads to symptoms such as abdominal pain, bloating, and flatulence [5]. In addition, food intake affects the intestinal microbiota, which is generally believed to play an important role in overall health [6].

### 1.3. Low-FODMAP Diets and Bakery Products

Diet represents a key role in IBS management because a large number of patients report food-related symptoms. A reduction in the named symptoms as well as overall life improvements in IBS patients were reported in multiple clinical studies investigating the effects of low-FODMAP diets [7]. FODMAPs are found naturally in many foods and food additives, including dairy products (milk), fruits (e.g., apples, pears, stone fruits), grains (wheat, rye), legumes, and vegetables (e.g., cauliflower, mushrooms, onions) [8,9].

Fructans and FOSs are the main FODMAPs in wheat-based products [10]. Importantly, wheat- and rye-based products are major food sources in Europe and therefore contribute significantly to FODMAP uptake. Austrian flour, which is analyzed in this study, is categorized by numbers indicating the ash content. The letters that precede these numbers indicate the grain type. For example, W700 represents wheat flour with an ash content ranging from 0.66% to 0.79% by dry mass.

During the baking process, fructans undergo partial degradation due to yeast invertase activity. The remaining fructans show a lower degree of polymerization (DP) than the initial flour product. Fermentation processes prior to baking generally allow for the partial conservation or degradation of FODMAPs [11]. Furthermore, methods for enzymatic degradation of FODMAPs in flour prior to fermentation have been reported recently [12]. 

The fermentation of flour–water mixtures to form sourdough is a long-practiced method used in traditional bread making. In typical wheat and rye sourdough fermentation, a complex microbial ecosystem comprises multiple heterofermentative lactic acid bacteria (LAB) and yeasts. LAB produce glucansucrases or fructansucrases, which convert FOSs to fructose, glucans, glucose, isomaltooligosaccharides, and sucrose [11]. Furthermore, heterofermentative LAB have been found to partially convert the released fructose molecules to mannitol [13]. Since mannitol is rapidly fermented by the gut microbiota, it should also be measured to provide accurate FODMAP identification [11]. In conclusion, LAB strongly increase the metabolic capacity of fermentation processes to convert and degrade FODMAP components [10].

In addition to degrading FODMAPs, sourdough fermentation may also generate FODMAPs through the digestion of complex carbohydrates. Thus, low-FODMAP baking also requires the presence of fructan- and mannitol-degrading organisms [11]. Additionally, arabinose is capable of reducing sucrose digestion by inhibiting sucrase activity in the small intestine. Arabinose itself is fermentable and slowly absorbed and thus has favorable effects on FODMAP-related symptoms [14].

### 1.4. Analysis of FODMAPs

High-pressure liquid chromatography (HPLC) used in combination with charged aerosol detection (CAD) or refractive index (RI) detection is one of the techniques used in the analysis of relevant FODMAPs, including FOSs, organic acids, polyols, and saccharides [15,16]. Due to a lack of standard substances for long-chain fructans and/or because of their insufficient purity, the approach of using universal detection with inverse gradient compensation (HPLC-CAD-IG) has been shown to be suitable [17,18]. Although FOSs, polyols and saccharides do not possess chromophores, which would make them optically detectable via diode array detectors (DADs), UV/Vis spectroscopy is a useful tool for ruling out the possible coelution of compounds with analytes of interest [19].

### 1.5. Contribution Beyond the State-of-Art

Recently, many papers successfully demonstrated the use of different bacterial strains for FODMAP reduction. Pure cultures in specialized fermentation media were used at optimal growing conditions (≥25 °C) to prove the capability of fructophilic lactic acid bacteria and Lactobacilli to degrade the oligo- and polymers to a significant extent [20,21]. Nonetheless, temperatures ≤ 10 °C are favorable in continuous production of sourdough as mold growth is promoted at common sourdough fermentation temperatures [22]. Additionally, the use of pure organisms is usually impossible for most bakeries and recreational bakers who mostly rely on their own continuously growing cultures, which are nourished with flours used in latter production, rather than on specialized growing media. Literature still lacks information regarding the relation between source material and continuously grown sourdough at temperatures ≤ 10 °C in terms of sourdough microbiome, fermentation byproducts, and FODMAP reduction. To the best of our knowledge, the formation of arabinose as a favorable fermentation byproduct has not been documented to date and should be more closely investigated.

Here, we report on a comprehensive approach to assessing the FOS content in bakery products and evaluating the flour used in their production as well as a microbiological approach to improving these products, as suggested by recent literature [23].

## 2. Materials and Methods

### 2.1. Materials and Reagents 

Standards of arabinose, fructose, glucose, maltose, and mannitol were purchased from Sigma-Aldrich (Schnelldorf, Germany). Kestopentaose, kestose, kestotetraose, FOSs (DP 2–8; P-FOS28), and inulin (DP 2–60; P-INUL) as well as the fructanase enzymes (EC 3.2.1.7 endo-inulinase and EC 3.2.1.80 exo-inulinase; E-FRMXLQ) were purchased from Megazyme (Wicklow, Ireland). Analytical-grade acetic acid, lactic acid, and citric acid were obtained as ready-to-use organic acid analysis standards from Bio-Rad (125–0586, Hercules, CA, USA). LC-MS-grade acetonitrile, ammonium acetate, sulfuric acid (96%), sodium hydroxide (50%), acetic acid (99–100%), and ammonia (25%) were obtained from VWR (Vienna, Austria). Flour (Table 3), bakery products (Table 2), and sourdough starter cultures were provided by Fischer Brot GmbH (Linz, Austria). Bio-Kornspitz, cottage bread, crusty bread, farm-baked bread, farmhouse bread, French bread, hard rolls, light bread, wheat-rye bread, and white bread were manufactured by Fischer Brot GmbH and are sold in retail stores as well as on the wholesale market in Austria. The sourdough starter cultures were purchased by Fischer Brot GmbH from Ernst BÖCKER GmbH & Co. KG (Minden/Westfalen, Germany).

### 2.2. Sample and Standard Preparation

Carbohydrate standards for isocratic separation were prepared at concentrations of 0.01, 0.05, 0.10, 0.50, 1.0, and 4.0 g/L in deionized water. The organic acid standard was prepared according to the manufacturer’s guidelines. Carbohydrate standards for inverse gradient separation on a Vanquish Flex system were prepared at concentrations of 0.010, 0.050, 0.10, 0.50, 1.0, and 4.0 g/L in 50% acetonitrile. Calibration curves were constructed using the measured peak areas. For measurements with CAD, no derivatization steps were required. The calculated carbohydrate content of the measured samples was corrected for dry matter content.

For carbohydrate and polymer extraction, 100 mg ± 5 mg flour, ground bakery products, or sourdough were weighed into centrifugation tubes and 500 µL deionized water and 500 µL acetonitrile were added. Extraction was performed at 40 °C for 120 min at 1000 rpm in a thermoshaker. After centrifugation at 17,000 × *g* at 25 °C for 5 min, 100 µL supernatant was transferred into a fresh centrifugation tube and vortexed after adding 450 µL acetonitrile and 450 µL deionized water. The precipitates were removed by repeated centrifugation under the abovementioned conditions. Approximately 850 µL supernatant was transferred into a screw-cap vial for HPLC measurement. The sourdough samples that were analyzed for their organic acid content were transferred into centrifugation tubes and centrifuged. The supernatant (100 µL) was transferred into a fresh centrifugation tube, followed by the addition of 500 µL acetonitrile and 400 µL deionized water.

FOSs and inulin were purified by fractionated collection to obtain FOS containing a sucrose unit (GFn) and pure fructose FOS (Fn) fractions to be used as standards in the subsequent analyses. The fractions were collected in centrifugation tubes, weighed, and analyzed for purity by HPLC-CAD after solvent evaporation at 60 °C in a SpeedVac. Hydrolysis was performed in a thermoshaker at 1150 rpm at 80 °C for 3 hours using fructanase enzymes to determine the fructose and glucose contents of each fraction.

For the analysis of the FOS content, the samples were extracted with water and centrifuged. The supernatant was utilized for additional sample pretreatment. To ensure that the relevant FOS content could be analyzed with this method, the supernatant was removed, and the residue was washed twice and subjected to enzymatic hydrolysis again. However, no DP 2–7 FOS or fructose were measured in the sample residue.

The sourdough samples were incubated at 4 °C, 10 °C, 20 °C, and 30 °C in a thermal incubator (Memmert IN260, Schwabach, Germany). Water was evaporated from the collected oligomer fractions with an Eppendorf 5301 concentrator (Eppendorf AG, Hamburg, Germany) attached to a KNF N840 Laboport (KNF DAC GmbH, Hamburg, Germany) oil-less vacuum pump. Equipment used for sourdough mixing and back-slopping was autoclaved before use.

### 2.3. Enzymatic Tests 

A 0.5 mol/L sodium acetate stock buffer solution for pH adjustment to 5.0 was used to stabilize the pH during hydrolysis. The fructanase stock solution was diluted to 1000 U/mL in deionized water. For the hydrolysis solutions, 100 mg ± 1 mg of flour, ground bakery products or sourdough were weighed into centrifugation tubes, and 800 µL deionized water, 100 µL stock buffer solution, and 100 µL enzymatic stock solution were added.

For the enzymatic test solutions, inulin (DP 2–60) was used. The amount of deionized water in the solutions was adjusted to obtain enzyme concentrations of 5, 20, 50, 100, 200, and 300 U/mL. Hydrolysis was performed in a thermoshaker at 1150 rpm at 80 °C for 1, 2, 3, 4, 5, and 6 hours under the same temperature and rotation conditions. After hydrolysis, the samples were centrifuged, and 100 µL supernatant was mixed with 500 µL acetonitrile and 400 µL deionized water. After vortexing, the samples were centrifuged again under the abovementioned conditions, and 850 µL supernatant was transferred into screw-cap vials for HPLC measurement.

### 2.4. Instrumentation

LC-MS-grade water (<0.055 μS/cm) from a Sartorius ultrapure water purification system (Sartorius, Göttingen, Germany) was used for sample preparation. Extractions were performed using a Thermal Shake lite (VWR, Vienna, Austria) thermoshaker. Dry matter measurements were performed using an MA 35 moisture analyzer (Sartorius, Göttingen, Germany).

FOS, fructose, glucose, maltose, mannitol, and sucrose analysis was performed on a Vanquish Flex UHPLC system (Thermo Fisher Scientific, Waltham, MA, USA). The system was equipped with a solvent degasser, binary pump, autosampler, and thermostatic column compartment and coupled to an Ultimate 3000 DAD-3000 DAD, followed by a Vanquish charged aerosol detector (Thermo Fisher Scientific, MA, USA). The compressed air gas flow rate was automatically regulated and monitored by the CAD device. The data collection rate was set to 2.0 Hz with a filter constant of 3.6 s. The power function for response and signal correction was set to 1.0. The evaporation tube temperature was maintained at 35 °C. For the inverse gradient conditions, a Thermo UltiMate 3000 HPG 3400 binary pump was connected via a viper capillary (50 µm × 950 mm) with a T-piece to the charged aerosol detector for backpressure consistency. The void volume differences between the analytical and inverse gradient flow paths were determined with a RefractoMax520 RI-Detector (ERC GmbH, Riemerling, Germany).

Chromatographic separation was achieved for carbohydrates and polymers with the Vanquish Flex system using two serially connected Waters Acquity UPLC BEH amide columns (130 Å, 1.7 µm, 2.1 mm × 150 mm) maintained at 35 °C. The autosampler was kept at 25 °C. A guard column (Acquity UPLC BEH Amide VanGuard precolumn, Milford, MA, USA) was used to protect the column from contamination by particles. Mobile phase A was 90% acetonitrile, and mobile phase B was 50% acetonitrile. Both mobile phases contained 10 mM NH4Ac adjusted to pH 9.00 with NH4OH (25%, NH3 basis). All buffer components were filtered through a 0.2 µm membrane filter. The gradient program used is shown in (Table 1). 

The autosampler needle was washed with 100 µL (20 µL/s) deionized water before and after each injection to prevent sample carryover between runs. The injection volume was 5 µL. The level of acetonitrile (70%) in the mobile phase reaching the charged aerosol detector was kept constant by setting the delay for the inverse gradient pump to 2.77 min. The flow rate was kept constant for both gradient pumps.

Organic acid, arabinose, fructose, glucose, and mannitol analysis was performed on an UltiMate 3000 UHPLC system (Thermo Fisher Scientific, Waltham, MA, USA) equipped with a solvent degasser, quaternary pump, autosampler, thermostatic column compartment, and RefractoMax520 RI detector. The data collection rate was set to 5.0 Hz at a rise time of 0.25 s. The temperature of the detector flow cell was kept at 35 °C. Data processing was carried out with Chromeleon 7.2.10 software (Thermo Fisher Scientific, MA, USA).

Chromatographic separation was achieved using a Bio-Rad Aminex HPX-87H column (9 µm, 7.8 mm × 300 mm) maintained at 30 °C. The autosampler was kept at 25 °C. A guard column (Micro-Guard Cation H Refill Cartridge, 30 mm × 4.6 mm, Hercules, CA, USA) was used to protect the column from particles. For isocratic separation at 0.6 ml/min, LC-MS-grade water (<0.055 μS/cm) was used. The mobile phase contained 0.010 mol/L H_2_SO_4_. The autosampler needle was washed with 100 µL (20 µL/s) mobile phase H_2_SO_4_ before and after each injection to prevent sample carryover between runs. The injection volume was 25 µL.

### 2.5. Sourdough Fermentation

The sourdough was fermented in triplicate batches. The so-called back-slopping technique was applied once a week by mixing 10 g of starter culture with 30 g deionized water and 30 g of the appropriate flour in a 500 mL Erlenmeyer flask. The subsequent fermentation was conducted in an incubator at 4, 10, 20, or 30 °C for 168 hours. After the elapsed time, 60 g of sourdough was removed and used for the subsequent analysis. The cultures were replenished with 30 g deionized water and 30 g flour, mixed thoroughly, and fermented for 168 hours. Thereafter, the procedure was repeated. The total experimental duration was 105 days. Experiments were aborted if mold-like growth was observed or an unpleasant smell was noted. The sourdough fermentation experiments were conducted with R960 rye flour (Haberfellner Mühle GmbH, Grieskirchen, Austria) and W700 wheat flour (Wiesbauer–Mühle GmbH, Mörschwang, Austria) cultures.

### 2.6. Next-Generation Sequencing (NGS) 

Next-generation sequencing was performed by Loop Genomics (San Jose, CA, USA). Loop Genomics maintains a cloud-based platform for processing raw short reads prepared with a LoopSeq kit into assembled synthetic long contigs. Within this pipeline, reads are trimmed to remove adapter sequences before they are demultiplexed based on their loop sample index, which groups reads by the sample from which they originated.

With enough short reads to cover the full length of a 16S molecule, it is possible to reassemble the original long sequence by linking the reads. Short reads whose sequences partially overlap can be linked through that shared sequence and then be arranged in the correct order to build the original 16S molecule sequence.

### 2.7. Statistics

Statistical analysis was performed with Chromeleon 7.2.10 (Thermo Fisher Scientific, Waltham, MA, USA) and Excel 365 (Microsoft, Redmond, WA, USA) using linear regression, intra- and interday repeatability, the unpaired Student’s *t*-test, and multivariable regression analysis. In the experiments, the mean is based on *n* = 9 individual samples. In this study, the results are presented as the means ± SDs and significance levels of *p* < 0.05 (c), *p* < 0.01 (b), and *p* < 0.0001 (a). Figures were prepared using CorelDraw 2019 (Corel Corporation, Ottawa, Ontario, Canada).

## 3. Results

### 3.1. FOS and Fructose Contents of Bread and Bakery Products 

Samples with low additive content (<1%) were chosen for these analyses. Crust-bread, wheat-rye-bread, white bread, and hard rolls contained rapeseed oil and yoghurt, which did not affect the FODMAP content. The fructose and DP 2–7 FOS content was determined in the supernatant after extraction and centrifugation in water by direct HPLC-DAD-IG-CAD detection. The results are shown in (Table 2 and Table 3) and are provided in order of increasing DP 2–7 FOS content.

Bakery products containing higher levels of wheat flour (hard rolls, light bread, white bread) showed the lowest FOS content (DP 2–7, ≤7.66 g/kg; DP > 7, ≤5.08 g/kg) of all tested samples. Lower FOS content for both DP 2–7 and DP > 7 were observed in the mixed wheat (≤40%) and rye (<30%) bakery products (crusty bread, farm-baked bread, French bread and wheat-rye bread) than in the cottage bread and farmhouse bread, which had a high proportion of rye flour (>50%). The Bio-Kornspitz sample, which consisted of mainly wheat (>45%) and less rye (<30%) flour, constituted an exception, as it exhibited a high FOS content (DP 2–7, 27.96 ± 0.45 g/kg; DP > 7; 22.07 ± 0.86 g/kg). Because the FOS and fructose contents of the bakery products varied, the flours used in those bakery products were analyzed.

### 3.2. FOS and Fructose Contents of Rye, Spelt and Wheat Flour 

Different types of rye, spelt, and wheat flour were evaluated for their long- and short-chain FOS and fructose contents (Table 2). Mineral content (rye: *r*^2^ = 0.92, *p* < 0.01; wheat: *r*^2^ = 0.87, *p* < 0.00008) and flour type (*r*^2^ = 0.86, *p* < 0.0000024) were identified as the two main factors influencing the FOS content in flour. Nevertheless, rye flour showed a higher long-chain FOS content at all milling degrees than wheat and spelt flour. Wheat flour of low mineral (W500–W700) content was comparable with spelt in terms of its fructose (−14.99%), short-chain FOS (−11.73%), and long-chain FOS (+7.15%) contents, whereas rye flour (R500–R960) showed higher fructose (6.9-fold) and short-chain (1.9-fold) and long-chain (4.2-fold) FOS content.

The fructose content in the whole-grain spelt flour was higher than that in the whole-grain wheat flour (+60.45%) but comparable with that in the whole-grain rye flour (+4.08%). The long-chain FOS content in the whole-grain rye and R2500 flour was much higher than that in the whole-grain wheat (3.2-fold; 4.1-fold) and whole-grain spelt (3.6-fold; 4.5-fold) flour.

The fructose (±9.10%; ±12.07%) and DP 2–7 (±5.73%; ±8.90%) and DP >7 (±0.59%; ±8.77%) FOS content of the W550 and R960 flours did not appear to vary by manufacturer. The deviations in the fructose (0.46 ± 0.16 g/kg) and DP 2–7 FOS (2.95 ± 0.25 g/kg) and DP >7 FOS (13.78 ± 1.51 g/kg) contents were more pronounced in the W700 flour than in the other flours but remained within a range that is typical for natural products. As their FOS content had been determined and did not appear to be dependent on the manufacturer, W700 and R960 flour, which came from different manufacturers, were selected for the sourdough fermentation trials.

### 3.3. FOS Content of Rye and Wheat Sourdough

Rye-based cultures fermented at 20 °C developed an unpleasant smell after 30 days; after 35 days, the fermentation process had to be stopped due to excessive mold growth. Rye fermentation at 30 °C was aborted after 7 days, as mold growth and olfactory issues became evident during that period. Results could not be obtained for wheat-based sourdough fermented at 20 °C and 30 °C, as both trials showed excessive mold growth within the first week of fermentation. For both rye and wheat sourdoughs, temperatures of 20 °C and 30 °C proved to be unsuitable for back-slopping once a week.

Mold growth was not observed in the rye- or wheat-based cultures fermented at 4 °C or 10 °C, and no olfactory abnormalities from these cultures were noted. In the course of the one-week fermentation period, the samples became more fluid, and a perceivably acidic smell was generated. Each of the biological replicates was analyzed for its fructose and FOS content in triplicate. The results, including the most relevant data points, are depicted in (Figure 1).

The results for the wheat sourdough replicates fermented at 4 °C and 10 °C are depicted in Figure 1A and Figure 1B, respectively. The fructose content in the wheat sourdough was equal to that in the rye sourdough throughout the fermentation trials, although the lower initial fructose content in the flour types used had to be considered in the comparison. However, at 10 °C, the content of DP >7 FOSs was significantly (*p <* 0.0001) decreased over the course of fermentation, whereas the level of short chain FOSs remained constant at <1 g/kg throughout the fermentation process. After an increase between day 28 and day 42, the DP 2–7 FOS content was detectable at similar concentrations in sourdough fermented at 4 °C.

After 105 days of fermentation, the microbiome in the wheat sourdough was capable of significant (*p <* 0.0001) reductions in the DP 2–7 FOS content, and the DP >7 FOS content compared with the initial content in the W700 flour at 4 °C (−60%, −99%) and 10 °C (−66%, −79%), respectively.

Rye sourdough samples fermented at 4 °C (Figure 1C) showed a decrease in DP 2–7 FOS content between day 42 (2.71 ± 0.82 g/kg) and day 84 (0.28 ± 0.07 g/kg). The content of long-chain DP >7 FOSs increased from day 28 (1.04 ± 0.11 g/kg) to day 84 (4.90 ± 0.14 g/kg) and then remained at the same level until day 105 (5.13 ± 0.14 g/kg). Additionally, changes in FOS content decreased noticeably throughout the whole fermentation period in the three independent cultures. Concerning fructose content, a significant (*p <* 0.0001) reduction compared with that in R960 flour (from 3.37 ± 0.31 g/kg to 0.57 ± 0.32 g/kg) was observed after 28 days of fermentation.

The rye flour fermentation results at 10 °C (Figure 1D) showed an increase in DP >7 FOS content between day 0 and day 28, a decrease from the initial level on day 42 (from 1.31 ± 0.02 g/kg to 0.67 ± 0.38 g/kg), and then stability until the end of the fermentation trials. A significant (*p <* 0.0001) decrease in DP 2–7 FOS content was observed after 42 days (−91.74%), but the decrease leveled off for the remaining fermentation period. The fructose content in the fermented rye sourdough at 10 °C was comparable with that in the rye sourdough fermented at 4 °C.

The early termination of the rye sourdough trials at 20 °C after 35 days prevented us from making reliable statements regarding the fructose and FOS content in these trials. Large standard deviations in the DP >7 FOS content were detected after 7 days. However, the deviations then began to stabilize, and the DP > 7 FOS content had increased in the samples that were analyzed after 21 and 35 days. The short-chain FOS content decreased significantly from day 21 (2.25 ± 0.39 g/kg) to day 35 (0.22 ± 0.28 g/kg) and was comparable to that in rye sourdough samples obtained at fermentation temperatures of 4 °C and 10 °C after 105 days, respectively. This trend also applied to the fructose content.

Compared with those in R960 flour, the fructose, DP 2–7 FOS, and DP >7 FOS content was significantly (*p <* 0.00001) decreased by the microbiome in the rye sourdough fermented for 105 days at 4 °C (−77%, −92%, −80%, respectively) and 10 °C (−77%, −89%, −98%).

### 3.4. Organic Acids Produced by Microorganisms in Rye and Wheat Sourdough

Organic acids are considered to greatly influence the taste and texture of a final bakery product. Therefore, wheat and rye sourdough were fermented at 4 °C and 10 °C for 105 days and evaluated for their lactate, acetate, and citrate contents during the fermentation process. The standard used for quantitation contained the sodium salt form of the organic acids. As the bond cations in sourdough were not analyzed and sulfuric acid was used in the mobile phase, it must be assumed that all acids were determined as H+ moieties. The standard used for quantitation contained malic acid, succinic acid, and oxalic acid as well, but none of those acids were found in the sourdough.

The values depicted in Figure 2 are the results of the analyses of three biological replicates at each temperature, measured in triplicate with HPLC-RI. The pH values of all analyzed sourdough samples were between 3.1 and 4.0. The microbiome of all sourdoughs under investigation showed increased organic acid production within the first 14 to 28 days. The total acidity after 105 days was markedly higher in rye sourdough fermented at 10 °C (Figure 2D; 57.92 g/L) than in rye sourdough fermented at 4 °C (Figure 2C; 38.51 g/L) and in wheat sourdough fermented at 4 °C (Figure 2A; 25.83 g/L) and 10 °C (Figure 2B; 39.39 g/L). Citrate was apparently not metabolized at 10 °C after 70 days of fermentation in rye or in wheat sourdough and therefore did not contribute to the total acid content.

The lactic acid to acetic acid ratio is well known for its importance in sourdough due to its sensorial influence on the taste and softness of the resultant bakery products. The ratio in rye and wheat (4:1) sourdough at 4 °C was remarkably higher in rye from day 56 to 105 than during other times during fermentation and in wheat (1:1) sourdough at 10 °C from day 70 to 105, independent of the absolute acidity of the sourdoughs. The acidity of the sourdoughs as well as the ratios of the three organic acids under evaluation had stabilized by 84 days of fermentation.

### 3.5. Fermentable Mono-, Disaccharides, and Polyols in Rye and Wheat Sourdough

The contents of the monosaccharides arabinose, glucose, maltose, and mannitol were analyzed throughout the fermentation period. As depicted in (Figure 3A,B), no arabinose was detected in wheat sourdough during the whole fermentation period, whereas maltose did not occur in rye sourdough at 10 °C (Figure 3D).

In wheat sourdough, peaks in glucose and maltose content at 4 °C and 10 °C (Figure 3A,B) were observable after 28 days. The same was true for mannitol at 10 °C. At 4 °C, mannitol was detectable for the first time on day 42 and remained at a constant level of 4.2 ± 0.6 g/kg thereafter. Wheat sourdough at 10 °C showed the lowest total monosaccharide content, 4.15 g/kg, on day 105. A direct comparison of rye sourdough at 4 °C and 10 °C revealed differences in glucose content: at 10 °C, the glucose content fell below the limit of detection, while levels of 44.6 ± 4.4 g/kg were maintained at 4 °C. The heavily fluctuating mannitol levels (27.4 ± 20.1 g/kg) in rye sourdough at 10 °C between day 0 and 63 levelled out to 26.9 ± 1.7 g/kg between day 84 and 105; in contrast, in rye sourdough at 4 °C, the mannitol content remained at 20.8 ± 2.2 g/kg throughout the whole fermentation period.

### 3.6. Analysis of the Microbiome in Rye and Wheat Sourdough

To identify the microorganisms responsible for the reduction in FOS content and the formation of arabinose and mannitol, NGS was performed. Although the same sourdough cultures were used for fermentation at 4 °C and 10 °C, the results revealed different microbiomes in the rye and wheat sourdough, as shown in (Figure 4).

An inconsistent microbiome was detected in the wheat sourdough at 4 °C (Figure 4A–C). While the wheat starter cultures (Figure 4A) incorporated >40% *Saccharomyces cerevisiae*, none of those cultures were detected on day 105 in the wheat sourdough at 4 °C. Although *Lactobacillus* sp. (68.3%) and *Fructilactobacillus sanfranciscensis* (30.8%) made up the majority of organisms on day 28, they were no longer detectable on day 105. At 10 °C, the microbiome of wheat sourdough proved to be more consistent than that at 4 °C after the substantial growth of *Acetobacter okinawensis* from day 28 until day 105 (Figure 4D,E). *Lactobacillus* sp. was present starting on day 0, with an average relative abundance of 13.47 ± 0.44%.

Lactobacilli represented the most abundant organisms in the rye sourdough (Figure 4F–J), including *Fructilactobacillus sanfranciscensis* (>20%) and *Lactobacillus* sp. (>18%). Additionally, in the rye sourdough at 10 °C (Figure 4I,J), *Acetobacter cerevisiae* (31.4%) multiplied until its abundance was almost equal to that of *Fructilactobacillus sanfranciscensis* (32.8%) on day 105.

## 4. Discussion

Clinical trials suggest that adverse effects in people suffering from IBS are worsened by the ingestion of dietary FODMAPs. Therefore, the identification and reduction of FODMAPs in staple foods represent an important step toward minimizing unfavorable effects. Comprehensive studies that connect the available products on the market with the flours used for production as well as propose potential methods of reducing FOS content are currently not available. The published research is limited to single aspects, such as the comparison of wheat and rye bread [24], the reduction of FODMAPs by microorganisms [25], or changes in the FODMAP profile in the malting process [26]. In this study, we demonstrate that long-term leavened sourdough at low temperatures exhibits significantly lower FODMAP levels compared with the original flour, and we identified and quantitated arabinose as a valuable byproduct of sourdough fermentation under these conditions. To our knowledge, no scientific publication quantitated the ratio of organic acids in sourdough fermented at 4 °C and 10 °C. Furthermore, we clarified the difference between rye and wheat sourdough, while proving a strong dependence of fermentation temperature on the degradation capability of long- and short-chain FOS. 

By investigating the relationships between bread, flour, and sourdough, both industrial and recreational bakers can be provided with the necessary information with which to develop bakery products with low FODMAP content.

To reach this goal, an advanced HPLC-DAD-IG-CAD method was developed and combined with a well-established HPLC-RI method to determine the FODMAP contents of bakery products and the flours used in their production. Furthermore, the products and organisms involved in back-slop sourdough fermentation were analyzed for their FODMAP content as well as for their fermentation byproducts.

### 4.1. Bread Consumption and IBS

It has been suggested that patients experiencing IBS should reduce their daily FOS intake to levels below 1.3 g/d, and a reduction in short-chain FOSs is especially advised [27]. Therefore, the FOS content of bread and bakery products was analyzed in order to estimate the need to decrease the contents of these compounds in typical products. In large-scale bakery production, leavening agents or stabilizers are added to dough. Therefore, the FOS content of these products cannot be compared directly with those of fermented products that are based on only wheat and water. However, the hard roll samples showed the lowest overall FOS levels (8.28 ± 0.3 g/kg), and the farmhouse bread had the highest levels (52.38 ± 1.34 g/kg). To comply with the guidelines of a low-FODMAP diet, >157 g/d hard rolls or >25 g/d farmhouse bread should not be consumed, and other FODMAP-containing foodstuffs must also be omitted. With 162.7 g/d of bread consumed per capita, this consumption recommendation is currently being surpassed in the European Union [28]. In our study, the FOS content increased in the order of wheat bread < mixed rye/wheat bread < rye bread. The Bio-Kornspitz sample represents an exception, as high FOS levels were detected despite its relatively low rye flour content. While sourdough was used for the fermentation of the other rye flour-based bread types in our study, the Bio-Kornspitz product was fermented by only yeast. It can be deduced that a yeast’s capability to decrease FOS levels is limited compared with that of sourdough. The FOS hydrolysis capability of yeast seems to be dependent on the strain used and to vary with invertase activity [29]. As all samples were fermented using industrial sourdough or yeast, an assessment of whether sourdough fermentation could be further improved was necessary.

### 4.2. Link between Raw Materials and FODMAP Reduction through Sourdough

Flour, the main component of all tested bakery products, was confirmed to be the key source of FOS. According to the literature, wheat flour has a similar FOS level to spelt flour, while the content in rye flour is higher [30]. R960 and W700 flours, representing the most common flour types used in Austria, were selected for the subsequent sourdough fermentation trials.

The fermentation process does not require complex equipment, which makes sourdough production easy to handle for both customers and manufacturers. The contents of DP 2–7 as well as DP >7 FOSs were significantly reduced in the rye and wheat sourdoughs at 4 °C and 10 °C compared with those in the flour used for fermentation. Wheat sourdough at 4 °C and rye sourdough at 10 °C showed the lowest overall FOS levels. Rye sourdough fermented at 10 °C proved to be most suitable for FOS and fructose degradation; this sourdough exhibited rapid adaptation after 42 days of fermentation, and its samples had the lowest standard deviations of all samples. Therefore, we concluded that this dough was the most stable in this study. In wheat sourdough at 10 °C, the hydrolysis of DP >7 FOSs increased during fermentation. The decrease in the long-chain FOS content of rye sourdough at 4 °C, along with its low DP 2–7 FOS level, could make this dough appropriate for use in bakery products with increased dietary fiber (inulin) content. Sourdough fermentation is an accepted method of FOS degradation, but this is achieved mostly through fermentation at higher temperatures [31]. No assertions about this process could be made for the high-temperature samples in our study, as we obtained only a single data point regarding fermentation at 30 °C. In general, higher fermentation temperatures led to unsatisfactory results from our back-slopping batch fermentation method.

However, it can be concluded that rye and wheat sourdoughs are capable of efficient FOS reduction, while yielding the desirable fermentation byproducts acetic acid, arabinose, citric acid, lactic acid, and maltitol. The recent literature suggests that the value of sourdough in influencing taste via fermentation byproducts is becoming increasingly important [32].

### 4.3. The Role of Fermentation Byproducts

The lactic acid to acetic acid ratio was 4:1 in both the rye and wheat sourdoughs at 4 °C and was 1:1 in the sourdoughs at 10 °C. The different ratios at the given temperatures suggest that distinct flavor profiles can be obtained depending on the fermentation temperature. The results suggest that the absolute content of both organic acids in bakery products could be mediated by fermentation time or by the sourdough proportion added. However, the ratio of organic acids in this study remained constant in both rye and wheat after 14 days at 4 °C and after 56 and 70 days at 10 °C, respectively. If dough proofing is conducted at the temperature used for fermentation, no significant change in those ratios should be expected, since proofing times are generally much shorter than fermentation times. Lactic acid and acetic acid are well known to occur in sourdough [33]. In addition to its positive effects on taste and texture, lactic acid was reported to reduce the acrylamide content of bread [34].

The citric acid content in the rye and wheat sourdoughs was approximately 20% at 4 °C. At 10 °C, the fermentation of citric acid stopped after day 70 in both sourdough types. No organisms specializing in the production of citric acid were detectable in our fermented sourdough. As an intermediate in the citric acid cycle, citric acid might have been excreted from the bacterial cytosol. The citric acid content of sourdough has not been reported in the literature to the best of our knowledge, and citric acid is not known to have any effects on fermented sourdough bread or bakery products. Considering that citric acid has a higher boiling point than acetic and lactic acid, citric acid should remain in bakery products if common baking temperatures (<250 °C) are used. The involatile nature of citric acid could inhibit the Maillard reaction in the bread crust, which is exposed to the highest temperatures. Presumably, this could lead to decreased acrylamide content, as observed with lactic acid.

Arabinose was detected in the rye sourdoughs fermented at 4 °C and 10 °C at 28.92 ± 10.98 g/kg. The ability to ferment arabinose has been reported in LAB [35]. Due to its inhibition of sucrase, arabinose has been reported to inhibit colitis by modulating the gut microbiota [36]. Colitis is defined as an infection of the large intestine and is related to symptoms that also occur in patients experiencing IBS [37]. Therefore, a high arabinose content in foods can be regarded as beneficial for reducing the inflammation that occurs in this patient group.

Mannitol was detected at higher concentrations (3.81 ± 2.44 g/kg) in rye sourdough than in wheat sourdough. The ability of lactobacilli to convert fructose to mannitol has been reported [11]. Mannitol cannot be metabolized by the human digestive system and is therefore used as a marker for intestinal permeability [38]. No adverse effects of high mannitol content on IBS symptoms are expected, but positive effects for patients with piriformis syndrome have been reported [38]. Mannitol is used in the food industry as an alternative sweetener to replace sucrose [39], providing a sweeter taste in bakery products [40]. Hence, rye sourdough could be used to alter the sweetness profile of the final bakery product.

The HPLC-RI device used for saccharide and polyol analysis was calibrated for erythritol, lactose, maltitol, sorbitol, sucrose, and xylitol. However, none of those compounds were detected in the sourdoughs.

### 4.4. Sourdough Microbiome

As suggested by the results, the microbiome of wheat sourdough seemed to be more capable of long-chain FOS degradation at a fermentation temperature of 4 °C than at 10 °C. Next-generation sequencing of the sourdough microbiome revealed the adaption of the sourdough microbiome to the flours used, with *Acetobacter cerevisiae*, *Acetobacter okinawensis*, *Fructilactobacillus sanfranciscensis*, and *Leuconostoc citreum* as the dominant organisms. These bacterial strains showed progressive growth, starting from low concentrations in the starter cultures. *Lactobacillus* sp. exhibited growth in each of the sourdoughs. However, due to the lack of a more precise classification, no further statements can be made regarding the specific metabolic processes performed by *Lactobacillus* sp. The growth of both *Fructilactobacillus sanfranciscensis* and *Leuconostoc citreum* was not expected in nonmodified sourdough (sourdough with artificially added cultures), but their presence is desirable in healthy sourdough. *L. sanfranciscensis* was identified as a key organism in high-quality sourdough fermented at 16–24 °C [41]. The growth of *L. sanfranciscensis* at temperatures ≤ 10 °C and the occurrence of *A. okinawensis* in low-FODMAP sourdough have not been reported to date. The growth of *L. citreum* was reported in wheat sourdough [42], and *L. citreum* has been used for the production of yeast-free wheat bread [43].

Although the results for the FOS, saccharide, and organic acid content indicate that desirable microorganisms showed ideal growth patterns, no assertion can be made regarding the bakery products produced with those sourdoughs. Future studies on the use of these products should be conducted to evaluate the sensorial and textural parameters of the final bakery products.

Fermenters are commonly used in the industrial production of yeast. For sourdough fermentation, the reproducibility of the results and the handling of the sourdough would benefit greatly from the use of specialized fermenters as well. To incorporate agitators and the automated feeding of flour–water mixtures into these experiments, the dry matter content of our original technique would need to be greatly reduced in order to provide better mixability and fluidity. Using fermenters, a temperature range between 2 °C and 10 °C should be tested with additional flour types in the future to clarify the influence of those parameters on the sourdough microbiome. Moreover, the addition of industrial yeast (*Saccharomyces cerevisiae*) to the sourdough at the dough stage should be tested in future studies, as the bread texture and sensorial parameters could possibly be improved by the use of yeast.

## 5. Conclusions

This study shows the feasibility of an easy-to-perform fermentation technique that can be industrially or domestically applied, as well as its benefits for the nutritional parameters of the sourdough produced. Back-slop-feeding on a daily basis is used in industry mostly due to the higher sourdough amounts required for production. Here, we suggest a method that primarily serves the cultivation of desired organisms rather than the purpose of producing large amounts of sourdough. 

As flour was identified as the main source of FOSs, the use of low-FOS flour can reduce the final FOS content in products prior to their further processing. We successfully created linkages between flour, the sourdough microbiome, and fermentation byproducts to sketch a coherent picture of this process. With the appropriate starter cultures, long-term fermentation without the growth of unwanted microbes is possible. We find that by varying fermentation temperatures and flour compositions, bakeries could grow different sourdough cultures and create blends with specific acid and sweetness profiles. We are convinced that the results of this study will contribute to a deeper understanding of sourdough fermentation and can lead to the development of modified products that provide additional benefits for people experiencing GI disorders, as well as for all other people.

## Figures and Tables

**Figure 1 foods-10-00894-f001:**
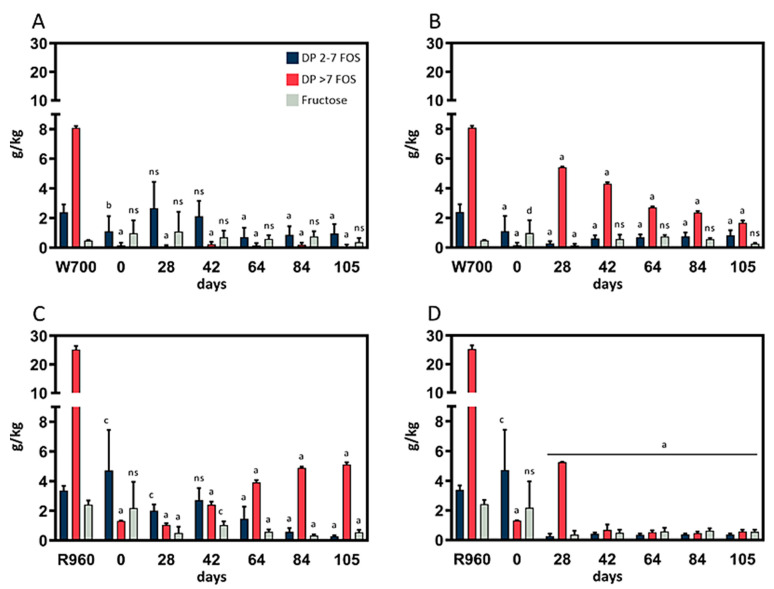
Fructose, DP 2–7 FOS, and DP >7 FOS content of sourdough (35 g water; 35 g flour). *p* < 0.05 (c), *p* < 0.01 (b), *p* < 0.0001 (a) and *p* > 0.05 (ns) indicate significant differences from the control (W700 or R960). (**A**) Wheat sourdough; 4 °C; W700. (**B**) Wheat sourdough; 10 °C; W700. (**C**) Rye sourdough; 4 °C; R960. (**D**) Rye sourdough; 10 °C; R960.

**Figure 2 foods-10-00894-f002:**
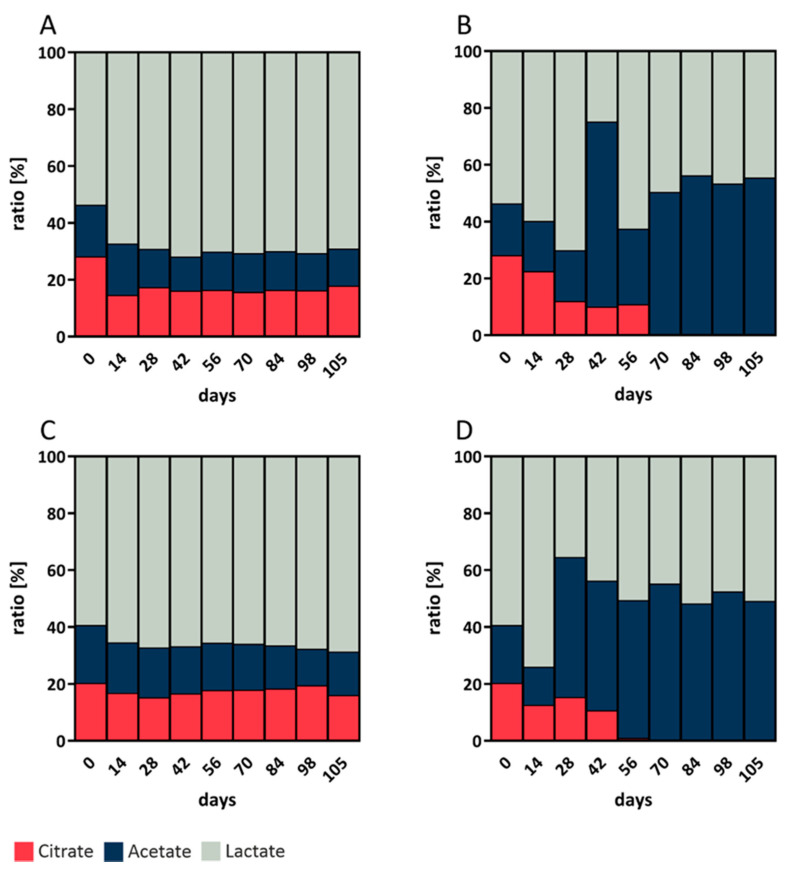
Acetic acid, citric acid, and lactic acid ratios of sourdough. (**A**) Wheat sourdough; 4 °C; W700. (**B**) Wheat sourdough; 10 °C; W700. (**C**) Rye sourdough; 4 °C; R960. (**D**) Rye sourdough; 10 °C; R960.

**Figure 3 foods-10-00894-f003:**
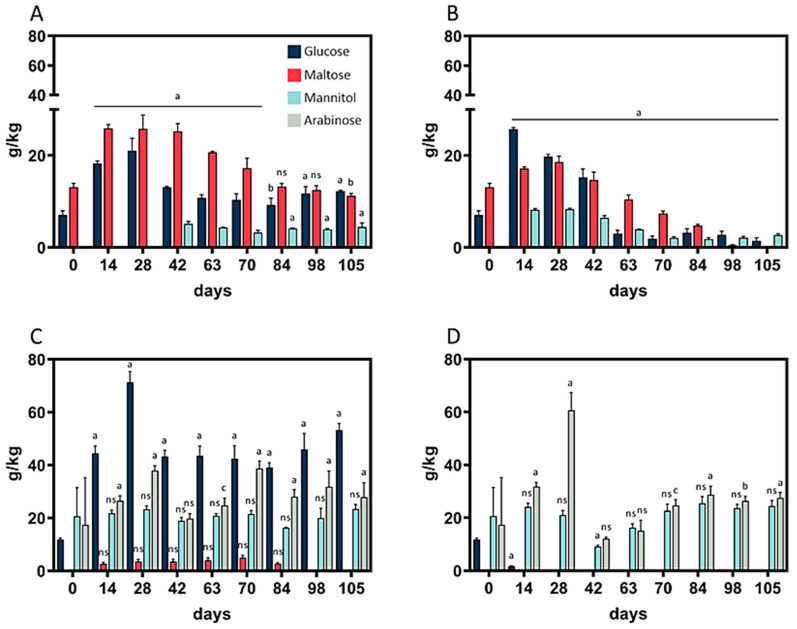
Arabinose, glucose, maltose, and mannitol contents of sourdough. (**A**) Wheat sourdough; 4 °C; W700. (**B**) Wheat sourdough; 10 °C; W700. (**C**) Rye sourdough; 4 °C; R960. (**D**) Rye sourdough; 10 °C; R960.

**Figure 4 foods-10-00894-f004:**
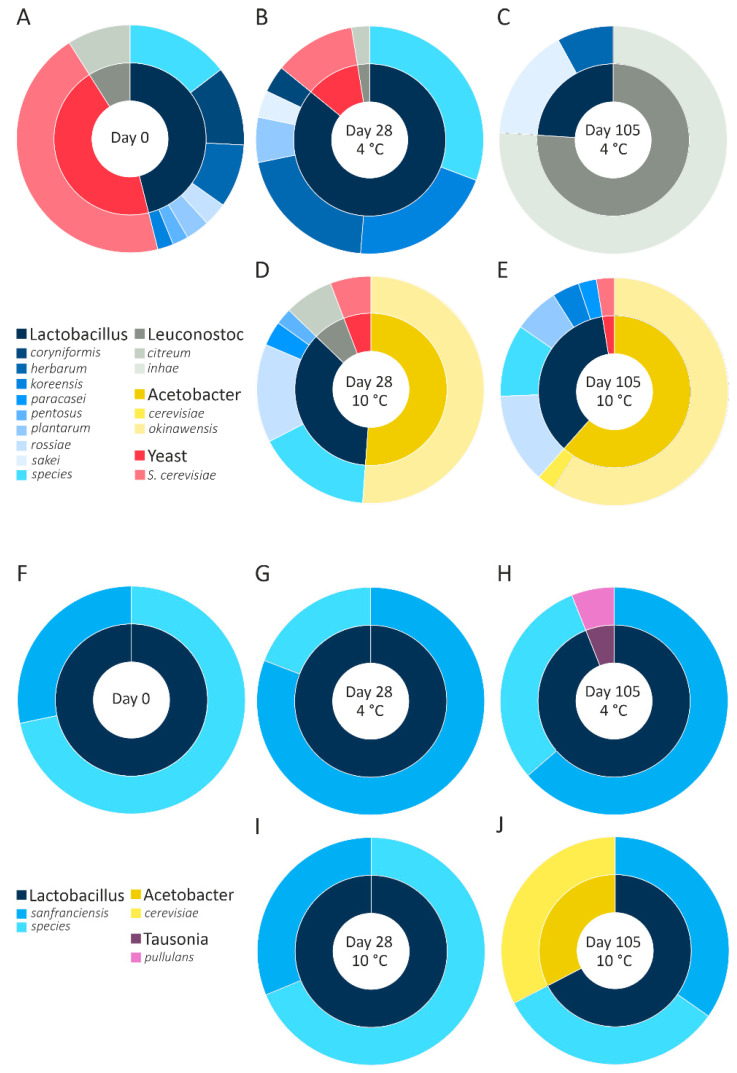
Microbiome of sourdough. Inner circle, genus; outer circle, species. (**A**) Wheat sourdough starter cultures. (**B**,**C**) Wheat sourdough, 4 °C. (**D**,**E**) Wheat sourdough, 10 °C. (**F**) Rye sourdough starter cultures. (**G**,**H**) Rye sourdough, 4 °C. (**I**,**J**) Rye sourdough, 10 °C.

**Table 1 foods-10-00894-t001:** Analytical and inverse gradient program.

Time (min)	Flow Rate (mL/min)	Analytical Gradient (Vanquish Pump)		Inverse Gradient (HPG 3400)	
		%A	%B	%A	%B
−10.000	0.23	99.0	1.0	1.0	99.0
0.000	0.23	99.0	1.0	1.0	99.0
5.000	0.23	99.0	1.0	1.0	99.0
35.000	0.23	1.0	99.0	99.0	1.0
45.000	0.23	1.0	99.0	99.0	1.0

**Table 2 foods-10-00894-t002:** Bakery products under study: types of sourdough used for dough proofing, flours used in production, FOS, and fructose content.

Product	Sourdough		Flour		FOS DP 2–7	FOS DP >7	Fructose
	Wheat	Rye	Wheat	Rye	g/kg		
hard roll	-	-	W550	-	5.09 ± 0.25	3.19 ± 0.34	1.91 ± 0.05
light bread	-	-	W550	-	5.50 ± 0.36	4.16 ± 0.18	1.34 ± 0.04
white bread	x	-	W550	-	7.66 ± 0.28	5.08 ± 0.50	2.58 ± 0.10
wheat-rye bread	x	x	W1600	R960	16.90 ± 1.36	13.12 ± 0.69	3.79 ± 0.08
French bread	-	-	W550	-	18.41 ± 1.54	16.87 ± 1.25	1.54 ± 0.05
farm-baked bread	-	x	W700;W1600	R960	20.04 ± 1.80	16.21 ± 0.71	3.82 ± 0.02
crusty bread	-	x	W1600	R960	22.33 ± 0.87	16.77 ± 1.80	5.56 ± 0.08
cottage bread	x	x	W700; W1600	R960	27.95 ± 1.28	20.13 ± 1.32	7.82 ± 0.03
Bio-Kornspitz	-	-	W550	mixed	27.96 ± 0.45	22.07 ± 0.86	5.89 ± 0.11
farmhouse bread	-	x	W1600	R960	28.08 ± 1.48	24.30 ± 1.20	3.78 ± 0.09

**Table 3 foods-10-00894-t003:** Flours under study: milling degree, FOS, and fructose content.

Flour		FOS DP 2–7	FOS DP >7	Fructose
Vendor	Type	g/kg		
Pfahndl	Eco flour	1.59 ± 0.03	7.78 ± 1.36	0.38 ± 0.03
Bio Mühle	W700	1.67 ± 0.60	8.64 ± 0.38	0.68 ± 0.15
Arnreiter	spelt	1.68 ± 0.13	6.07 ± 0.20	0.30 ± 0.02
Wiesbauer	W700	1.70 ± 0.13	7.91 ± 0.27	0.35 ± 0.05
Polsterer	W550	1.71 ± 0.25	7.77 ± 0.16	0.25 ± 0.02
Haberfellner	W550	1.76 ± 0.07	7.68 ± 0.25	0.27 ± 0.03
Haberfellner	R500	1.90 ± 0.27	19.28 ± 0.99	0.81 ± 0.29
Haberfellner	W500	1.91 ± 0.13	7.75 ± 0.08	0.30 ± 0.03
Klinger	W700	2.39 ± 0.53	8.09 ± 0.12	0.48 ± 0.03
Good Mills	W700	2.67 ± 0.28	8.66 ± 0.44	0.34 ± 0.06
Bio Mühle	spelt	2.68 ± 0.37	8.92 ± 0.17	0.59 ± 0.05
Haberfellner	W1600	2.77 ± 0.49	12.72 ± 1.17	1.24 ± 0.11
Klinger	W1600	3.12 ± 0.21	14.85 ± 0.10	1.26 ± 0.09
Klinger	R960	3.37 ± 0.31	25.20 ± 1.32	2.42 ± 0.28
Premium-Bio Mühle	R960	3.39 ± 0.66	23.62 ± 0.83	1.94 ± 0.11
Haberfellner	wheat whole grain	3.82 ± 1.28	13.05 ± 0.69	2.42 ± 0.20
Haberfellner	R960	3.93 ±0.33	28.05 ±0.48	2.41 ± 0.25
Arnreiter	spelt whole grain	3.94 ± 0.19	11.34 ±0.29	1.51 ± 0.13
Haberfellner	rye whole grain	4.28 ±0.24	29.34 ± 1.11	2.52 ± 0.28
Haberfellner	R2500	5.44 ± 2.12	40.06 ± 1.63	4.86 ± 0.92

## Data Availability

The data presented in this study are available on request from the corresponding author. The data are not publicly available due to privacy restrictions.

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
