# Peer review of "FODMAP Fingerprinting of Bakery Products and Sourdoughs: Quantitative Assessment and Content Reduction through Fermentation"

_foods, 2021, doi:10.3390/foods10040894_

Round 1
Reviewer 1 Report
The paper deals with a topic undoubtedly worth of investigation and is well presented. The manuscript is rather long and I warmly suggest merging the sections Results and Discussion in order to avoid repetitions. Similarly, the introduction may be shortened and some info may be used to discuss data.
I just have the following few remarks.
First of all the new nomenclature for lactobacilli should be adopted and maybe the British version of the word ‘mold’ is to be preferred.
I think authors should have a look of this recent paper: ‘Determination of FODMAP contents of common wheat and rye breads and the effects of processing on the final contents’ by Marcus Schmidt and Elisabeth Sciurba.
The second sentence of the abstract (lines 16-20) should be rephrased to clarify the concept and species should be in italicus
Line 230: Starter composition and source must be revealed. Moreover, I wonder if conditions for back-slopping were controlled. I mean: equipments were sterile? And the water? And the flours? Authors should clarify.
Lines 260-270: Please delete this intro, it does not have any sense since tables and figures are commented later on.
Table 2: Why ‘types of microorganism used for dough proofing’?
Fig. 2: Absolute concentrations provide a more useful info. Please change the figure.
Fig. 4: maybe the temperature may be inserted in the circle below the day.
Lines 431-435: Please rephrase. Obviously Saccharomyces cerevisiae could never have retrieved and it is hard to follow the authors’ speculation without indications about the starter's composition…
Line 452: ‘yeast and rye bread?
Author Response
The paper deals with a topic undoubtedly worth of investigation and is well presented. The manuscript is rather long and I warmly suggest merging the sections Results and Discussion in order to avoid repetitions. Similarly, the introduction may be shortened and some info may be used to discuss data. Reading the following fragments of the manuscript, it is hard to resist the impression that the authors have studied the bibliography rather cursily and approach their experiences partly uncritically and partly with unreasonable enthusiasm.
We thank the reviewer for critically reading our manuscript. Indeed, there was room for improvement and clarification, accordingly, we tried to address the raised points properly. We kept the seperate results and discussion sections, however, we shortened both sections accordingly.
- First of all the new nomenclature for lactobacilli should be adopted and maybe the British version of the word ‘mold’ is to be preferred.
Lactobacillus sanfranciscensis was changed to Fructilactobacillus sanfranciscensis. Mold has been changed to mould.
- I think authors should have a look of this recent paper: ‘Determination of FODMAP contents of common wheat and rye breads and the effects of processing on the final contents’ by Marcus Schmidt and Elisabeth Sciurba.
We carefully read the suggested paper and included it into our bibliography (Line 56).
- The second sentence of the abstract (lines 16-20) should be rephrased to clarify the concept and species should be in italicus
Species names were italicized, and the abovementioned section was rephrased.
- Line 230: Starter composition and source must be revealed. Moreover, I wonder if conditions for back-slopping were controlled. I mean: equipments were sterile? And the water? And the flours? Authors should clarify.
The composition of the starter cultures is depicted in Figure 4A. The source is revealed in Line 125-127: “The sourdough starter cultures were purchased by Fischer Brot GmbH from Ernst BÖCKER GmbH & Co. KG (Minden/Westfalen, Germany).”. Equipment was sterilized before use (added: Line 164-165), LC-MS grade water was used (Line 160-162) and the flours were not treated before use.
- Lines 260-270: Please delete this intro, it does not have any sense since tables and figures are commented later on.
We deleted the intro and rearranged the tables and figures.
- Table 2: Why ‘types of microorganism used for dough proofing’?
Thanks for the remark. It should mean ‘types of sourdough used for dough proofing’ and was corrected accordingly.
- 2: Absolute concentrations provide a more useful info. Please change the figure.
We decided to use the relative instead of absolute concentrations for better comparability. In respect of flavor profile, we consider the relative values as a more significant information.
- 4: maybe the temperature may be inserted in the circle below the day.
The temperature values were added as suggested.
- Lines 431-435: Please rephrase. Obviously Saccharomyces cerevisiae could never have retrieved and it is hard to follow the authors’ speculation without indications about the starter's composition…
The starter composition is indicated as day 0 culture in Figure 4A. Indication added in Line 435.
- Line 452: ‘yeast and rye bread?
Thanks for the remark. We corrected it to “wheat and rye bread”.
Reviewer 2 Report
The paper “FODMAP Fingerprinting of Bakery Products and Sourdoughs: Quantitative Assessment and Content Reduction Through Fermentation” reports the results of a study focused on reduction of FODMAP in bakery products. Bakery products and flours are first analysed, experiments with different fermentation conditions and raw materials are then done, and finally NGS is made to understand the link between the obtained results and the sourdough microbiome.
The topic is indeed relevant and of interest to the scientific community, as well as to the consumers, as it is known that FODMAPs play a major role in gastrointestinal disorders.
Some minor suggestions to make it fully publishable:
Abstract: Please, use italics for starter names.
Introduction: It is quite long, however the decision to organise it into sub-paragraph makes it very readable. It well introduces the reader to the experimental part.
Materials and Methods: this part is generally fine. I only suggest adjusting the title of paragraph 2.3 (line 159) to its content or split the information included in it among the other paragraphs.
Table1: check the font. I’m not sure it is Palatino Linotype.
Lines 225 and 226: use the subscript for H2SO4.
Results: a general suggestion is to have a paragraph (3.1) also for the first part of the section (lines 260-278).
Lines 280-282: it is not clear if you are aware of the possible addition of additives to some of the bakery products you analysed. If so, it might be interesting to add this information in the text.
I noticed to put Figure XX within brackets, such as at line 331, 334, 381, 426, etc. Please, check Journal instructions and make sure brackets are mandatory or otherwise.
Lines 433-434: one line.
Discussion:
Lines 483-484: did I understand correctly that your hypothesis about the fact that “The Bio-Kornspitz sample represents an exception, as high FOS levels were detected despite its relatively low rye flour content.” is that it was fermented with yeasts?
Lines 557 and 576: please, use numbers for these references.
Lines 624-628: please, make sure you have to keep this part in the way you wrote it.
References: the layout does not meet the instructions of the Journal. Please, amend them.
Author Response
The paper “FODMAP Fingerprinting of Bakery Products and Sourdoughs: Quantitative Assessment and Content Reduction Through Fermentation” reports the results of a study focused on reduction of FODMAP in bakery products. Bakery products and flours are first analysed, experiments with different fermentation conditions and raw materials are then done, and finally NGS is made to understand the link between the obtained results and the sourdough microbiome.
The topic is indeed relevant and of interest to the scientific community, as well as to the consumers, as it is known that FODMAPs play a major role in gastrointestinal disorders.
We thank the reviewer for critically reading the manuscript and for the positive feedback. The manuscript was carefully revised.
- Abstract: Please, use italics for starter names.
We italicized the starter names and changed Lactobacillus sanfranciscensis to the new nomenclature Fructilactobacillus sanfranciscensis.
- Introduction: It is quite long, however the decision to organise it into sub-paragraph makes it very readable. It well introduces the reader to the experimental part.
We sincerely thank the reviewer for the positive feedback.
- Materials and Methods: this part is generally fine. I only suggest adjusting the title of paragraph 2.3 (line 159) to its content or split the information included in it among the other paragraphs.
Contents of the first paragraph were split among the other paragraphs.
- Table1: check the font. I’m not sure it is Palatino Linotype.
Thank you for the remark. We changed the font to Palatino Linotype.
- Lines 225 and 226: use the subscript for H2SO4.
Subscript applied for atom count.
- Results: a general suggestion is to have a paragraph (3.1) also for the first part of the section (lines 260-278).
The paragraph was removed as the figures and tables are commented in the next paragraphs (3.1 and 3.3).
- Lines 280-282: it is not clear if you are aware of the possible addition of additives to some of the bakery products you analysed. If so, it might be interesting to add this information in the text.
We added the information on additives.
- I noticed to put Figure XX within brackets, such as at line 331, 334, 381, 426, etc. Please, check Journal instructions and make sure brackets are mandatory or otherwise.
We removed the brackets according to journal instructions.
- Lines 433-434: one line.
We corrected that.
- Lines 483-484: did I understand correctly that your hypothesis about the fact that “The Bio-Kornspitz sample represents an exception, as high FOS levels were detected despite its relatively low rye flour content.” is that it was fermented with yeasts?
We hypothesize that the lack of sourdough cultures in the fermentation process was responsible for high FOS content in the final product. We therefore conclude that the yeast used for fermentation seems to be less efficient for FODMAP degradation.
- Lines 557 and 576: please, use numbers for these references.
We corrected that.
- Lines 624-628: please, make sure you have to keep this part in the way you wrote it.
We removed this part.
- References: the layout does not meet the instructions of the Journal. Please, amend them.
References were amended to the latest instructions of the journal.